# Malignant Transformation of Proliferative Verrucous Leukoplakia: A Description of the Clinical Oral Characteristics of These Squamous Cell Carcinomas

**DOI:** 10.3390/cancers17071199

**Published:** 2025-04-01

**Authors:** Jose Bagan, Judith Murillo, Jose M. Sanchis, David Hervás, Leticia Bagan

**Affiliations:** 1Medicina Bucal Unit, Stomatology Department, Valencia University, 46010 Valencia, Spain; leticia.bagan@uv.es; 2Precancer and Oral Cancer Research Group, Valencia University, 46010 Valencia, Spain; 3Service of Stomatology and Maxillofacial Surgery, University General Hospital, 46014 Valencia, Spain; jumucor@alumni.uv.es (J.M.); jose.m.sanchis@uv.es (J.M.S.); 4Department of Applied Statistics and Operational Research, and Quality, Universitat Politècnica de València, 46022 Valencia, Spain; daherma@eio.upv.es

**Keywords:** proliferative verrucous leukoplakia, oral squamous cell carcinoma, T stage

## Abstract

Proliferative verrucous leukoplakia (PVL) is a potentially malignant disorder with a high tendency to develop oral squamous cell carcinoma. Many authors agree that PVL is most at risk for oral squamous cell carcinoma (OSCC). The current study compares the clinical characteristics of two groups of patients: group 1 comprised OSCC with PVL, and Group 2 with only conventional OSCC. We tried to prove that OSCC cases with PVL have clinical characteristics different from those of conventional OSCC cases (group 2). Also, in group 1, the patients are diagnosed in an early T stage of the disease.

## 1. Introduction

Proliferative verrucous leukoplakia (PVL) is a rare disease, described by Hansen et al. in 1985 [1]. It manifests as plaques of leukoplakia that are initially homogeneous but become multifocal, sometimes affecting a large part of the oral mucosa, frequently acquiring a verrucous character [1,2]. According to Warnakulasuriya et al., PVL is a potentially malignant disorder with a high tendency to develop oral squamous cell carcinoma [3]. Other authors agree that PVL is most at risk for oral squamous cell carcinoma (OSCC) [4,5].

PVL has a high likelihood of recurrence after treatment with CO_2_ lasers or photodynamic therapy, its most common treatment modalities [6,7,8,9,10,11,12]. A meta-analysis yielded a recurrence rate of 67.2% (95% CI 48.3–81.8), without publication bias [13].

The most significant issue with PVL is its high rate of transformation into oral squamous cell carcinoma (43.87–65.8%), making PVL the oral disorder with the highest propensity for malignancy [14]. Other authors reported similar rates of malignant transformation [15]. However, many of the articles published on PVL are isolated clinical cases, with few large series with patients and their clinical characteristics, as well as their transformation into oral squamous cell carcinoma.

One of the first series to be published, after that of Hansen et al. [1] was that of Silverman and Gorsky with 54 cases [16]. In a mean time of 7.7 years, 70.3% of the patients developed a squamous cell carcinoma at a proliferative verrucous leukoplakia site, most frequently the gingiva and tongue. Twenty-one of the patients with proliferative verrucous leukoplakia died of proliferative verrucous leukoplakia-associated carcinoma. Zakrzewska et al. [17] reported 10 cases of PVL that all ended in OSCC. The lesions affected various intraoral sites, but the most commonly affected sites were the mandibular alveolar mucosa or gingiva (eight patients) and the buccal mucosa (eight patients). Lesions in a particular anatomical site were bilateral in nine of the ten patients. In a systematic review by Pentenero et al. [18] they selected 20 articles and found malignant transformation rates ranging from 40–100% of cases, with the most frequent location of OSCC in patients with PVL being the mucosa-gingival alveolar in most the articles.

In a very recent article with 170 cases of PVL, the risk factors for PVL oral squamous cell carcinoma were identified as including sex, the location of the lesions, the clinical presentation, not being a smoker, and oral epithelial dysplasia (OED) [19].

The OSCC occurring in patients with PVL (OSCC-PVL) differs clinically from typical OSCC not preceded by PVL (OSCC-noPVL). Faustino et al. evaluated the prognostic outcomes of OSCC-PVL in terms of recurrence, new primary tumors, metastasis, and survival outcomes and reported that it has better clinical outcomes than OSCC-noPVL [20]. Similarly, Gonzalez-Moles et al. reported that OSCC-PVL has more favorable prognoses than OSCC-noPVL, particularly in the mortality rate [21]. However, Faustino et al. found little information on the key prognostic outcomes of OSCC-PVL [20]; therefore, studies need to compare OSCC-PVL with OSCC-noPVL.

In a recent preliminary study involving eight cases of PVL that progressed to oral squamous cell carcinoma (OSCC-PVL) and ten classical OSCC cases (OSCC-noPVL), we found that the clinical and evolutionary characteristics of these oral squamous cell carcinomas differed. We also discovered that OSCC-PVL patients had lower expression of cancer-related genes. Hypermethylation in the promoter region of many genes was noted, suggesting that DNA methylation serves as a regulatory mechanism [22].

The current study compares the clinical characteristics of large groups with OSCC-PVL and OSCC-noPVL, to validate our preliminary results and demonstrate that OSCC-PVL has different clinical behavior to that of OSCC-noPVL.

## 2. Materials and Methods

This retrospective case control study enrolled patients diagnosed with OSCC, seen between 2005 and August 2024, who visited the University General Hospital of Valencia, Spain, and or a private clinic nearby.

The subjects comprised 50 patients with OSCC preceded by proliferative verrucous leukoplakia lesions (OSCC-PVL) and 90 with oral squamous cell carcinoma who had never experienced PVL lesions (OSCC-noPVL). The study was approved by the clinical trials and drug committee of the University General Hospital on 24 February 2023 (reference 10-2023).

This study is a retrospective case control study, so we included all available samples for the OSCC-PVL group. Regarding the number of samples for the OSCC-noPVL group, a power analysis was performed using Monte Carlo simulations to assess the best ratio of OSCC-noPVL to OSCC-PVL samples that would improve the statistical power without dramatically increasing the sample size. The results of the simulations concluded that a 2:1 ratio was the most cost-effective approach.

We recorded the patients’ ages and genders, the locations of the oral squamous cell carcinomas in the oral cavity, and the clinical types, distinguishing between erythroleukoplakia, ulcerated, exophytic, and mixed. The inclusion criteria for the patients in the OSCC-PVL group were those published by Cerero et al. [23]. The criteria are: Major criteria: (a) Leukoplakia lesion with more than two different oral sites; (b) Verrucous appearance; (c) Spreading or engrossing during the disease development; (d) Presence of recurrence in a previously treated area; (e) Histopathological test: oral epithelial hyperkeratosis to verrucous hyperplasia, verrucous carcinoma, or squamous cell carcinoma, whether in situ or infiltrating. Minor criteria: (a) Oral leukoplakia lesion that occupies at least 3 cm when adding all the affected areas; (b) Female patient; (c) Non-smoker regardless of gender; (d) More than 5 years evolution. Diagnostic criteria: Three major criteria (one of which must include the evolution of the histopathological lesions) or two major criteria (one of which must include the evolution of the histopathological lesions) + two minor criteria.

OSCC-noPVL consisted of OSCC patients who had not had PVL lesions. All patients with OSCC were diagnosed by biopsying the oral lesion.

Considering that the title of the article is Malignant Transformation of Proliferative Verrucous Leukoplakia, a description of the clinical oral characteristics of these squamous cell carcinomas, and that the sole objective of the article was to compare the clinical characteristics in the oral cavity of large groups with OSCC-PVL and OSCC-noPVL, we indicate the value of the tumor size (T) considering initial sizes T1 and T2, and on the other hand the advanced sizes (T3 and T4). T1 is considered to be an OSCC lesion size equal to or less than 2 cm, T2 being 2–4 cm, T3 if it was greater than 4 cm and T4 if it infiltrated neighboring structures [24].

The data were summarized using the mean (standard deviation) and median (1st and 3rd quartiles) for numerical variables and absolute (relative) frequencies for categorical variables. We compared quantitative and qualitative variables between both groups to identify significant differences We used the Wilcoxon–Mann–Whitney test to compare quantitative variables. For qualitative variables, we performed a contingency analysis using the Chi-squared test. For these analyses, we set the level of significance at *p* < 0.05.

Finally, a Bayesian multivariable logistic regression model was adjusted to assess whether the different clinical variables discriminated between the two groups. The uncertainty of the estimates was assessed by estimating the 95% credible intervals, and the probability of direction (pd) was used as an index of the existence of the effect. Regularizing priors, N(0, 3) were used to improve the robustness and reliability of our model results. The model’s performance was assessed by estimating the area under the receiver operating characteristic curve (AUROC). Internal validation of the AUROC was performed using 10-fold cross-validation. All statistical analyses were performed using R (ver. 4.4.1) and the R packages brms (ver. 2.22.0) and bayestestR (ver. 0.15.0).

## 3. Results

The mean age of the 140 patients with OSCC was 68.48 ± 11.42 (range 40–97) years. Of the 140 patients, 74 (52.9%) were male and 66 (47.1%) female. Table 1 shows the values of the different clinical variables analyzed. Comparing the two groups, the patients with OSCC-PVL were younger than those with OSCC-noPVL, but the difference was not significant (Mann–Whitney U = 1940, *p* > 0.05). There was a significant difference in gender; OSCC-PVL was much more common in women.

Considering the location of the oral squamous cell carcinoma in the mouth, half of the oral squamous cell carcinomas in OSCC-PVL were located in the gingiva, with a lower percentage in OSCC-noPVL. The erythroleukoplakia type (Figure 1) was much more frequent in OSCC-PVL, while the more clinically aggressive ulcerated form (Figure 2) predominated in OSCC-noPVL. Finally, on comparing the differences between the initial (T1 and T2) and advanced (T3 and T4) sizes of oral squamous cell carcinoma in the two groups, OSCC-PVL group was more frequent in the initial T sizes than the OSCC-noPVL.

The Bayesian logistic regression model showed that all the variables assessed differed between the OSCC-PVL and OSCC-noPVL groups. Specifically, being female was associated with OSCC-PVL (OR = 5.1, 95% CrI [1.9, 14.0], probability of effect >99.99%). The advanced stage was associated with the oral squamous cell carcinoma group (OR for OSCC-PVL = 0.34, 95% CrI [0.11, 0.91], 98.5% probability of effect). All clinical types except erythroleukoplakia were associated with the oral squamous cell carcinoma group (ORs for OSCC-PVL = 0.21, 0.35, and 0.15, for ulcerative, exophytic, and mixed, respectively, probabilities of effect of 99.3, 92.4, and 98.0%). Finally, locations at the floor of the mouth, tongue, and palate were also associated with the oral squamous cell carcinoma group (ORs for OSCC-PVL = 0.03, 0.31, and 0.13, respectively, probabilities of effect 99.3, 98.8, and 97.4%) (Table 2).

Figure 3 assesses the performance of the model by estimating the ROC curve for discriminating between OSCC-noPVL and OSCC-PVL patients. The model performed well with AUC = 0.84. Interval validation using 10-fold cross-validation yielded a validated AUC = 0.81, indicating good out-of-sample performance of the model.

## 4. Discussion

Oral cancer represents nearly 3% of new cancer cases in the United States, with incidence rates rising over the past decade. The overall incidence of lip and oral cavity cancers is approximately 4.1 cases per 100,000 people; however, there is significant variation worldwide, with higher rates observed in Asian countries [25].

A recent retrospective study analyzed the epidemiological and clinical characteristics of OSCC in 243 patients from Galicia, Spain. The average patient age was 67 years, and the majority were male (69.5%) [26].

In a study of classical OSCC (OSCC-noPVL), Saldivia-Siracusa et al. reported that 58 (54.2%) of the patients were men, with a mean age of 60.69 years; 49 (45.8%) and 39 (36.5%) patients had histories of tobacco and alcohol use, respectively [27].

In our series of 50 patients (group OSCC_PVL) the average age was 66.62, very similar to the figures of Amezaga-Fernandez et al. [26] and our OSCC-noPVL group and slightly higher than that of other authors [27].

With regard to the gender of the patients in our OSCC_PVL group, the frequency in women was much higher than in men, unlike what happened in the group not associated with PVL. In the Galician OSCC study, men also predominated, as in other studies such as that of Saldivia-Siracusa et al. [27] and Daroit et al. [28]. This shows that those patients with LVP who develop OSCC affect women much more, probably the justification for this is that men smoke more than women, and in our OSCC_LVP group women predominated.

In this sense, we point out that we did indeed find statistical significance between our two oral squamous cell carcinoma groups when considering whether they used tobacco, this habit being much more frequent in the second group (OSCC_noPVL) (Table 1). The lower frequency of tobacco use in PVLs has already been described by several authors such as Silverman et al. [16] as well as in the systematic review and meta-analysis by Pentenero et al. [18]. On the contrary, Kovalski et al. [29] reported that in OSCC-noPVL, tobacco consumption *(p* = 0.003) and alcohol intake (*p* = 0.02) were significantly greater in males than in females [29]. This was confirmed by Amezaga-Fernandez et al., who found that 45.9% of patients with OSCC were smokers and 58.6% consumed alcohol [26].

PVL is a potentially malignant disorder that is enigmatic due to our ignorance of its etiopathogenesis and therapeutic management [1]. No viral agents have been found to be involved in Upadhyaya et al. [30]. Recent research on the origins of PVL has concentrated on genetic analyses involving methylation, transcriptomics, and microbiota. Morandi et al. [31] evaluated the diagnostic value of methylation levels in a set of 18 genes using bisulfite next-generation sequencing, focusing on OSCC and other potentially malignant disorders, such as lichen planus and PVL. Their data highlight the importance of CpG islands’ location and accurate estimation of DNA methylation levels for an exact early diagnosis of OSCC [31].

Okoturo et al. [32] performed whole exome sequencing of five cases of OSCC-PVL, using paired blood samples to identify somatic mutations prevalent in the tumors. They discovered that, unlike classical OSCC, OSCC-PVL had rare TP53 mutations and altered patterns of *PIK3CA* and *NOTCH1* mutations. They concluded that the two groups have differences in mutation and methylation profiles.

Some authors report that the risk of malignancy associated with PVL is approximately 50%. Iocca et al. [33] defines the malignant transformation (MT) rate of oral potentially malignant disorders (OPMD) and the risk of developing mild versus moderate/severe oral dysplasia. With regard to PVL, they found malignant transformation figures of 49.55% (99% CI: 26.7–72.4%). However, Villa et al. [34] reported malignant transformation in 71.4% of patients with PVL after a median of 37 months from the initial visit; erythroleukoplakia underwent MT in 100% of cases. Ramos-Garcia et al. [15] stated that the pooled proportion of MT in PVL was 43.87%: females (64.02%) and males (35.98%). The most frequent sites of PVL were the gingiva (39.6%) and buccal mucosa (21.6%). No conclusions were drawn regarding MT concerning sex or age distribution or tobacco or alcohol consumption. The gingiva was the most common site of MT (39.9%).

In a recent preliminary study [22], we examined eight cases of PVL that progressed to oral squamous cell carcinoma (OSCC-PVL) and 10 classical OSCC cases (OSCC-noPVL) and found that the clinical and evolutionary characteristics of these oral squamous cell carcinomas differed. OSCC-PVL patients had lower expression of cancer-related genes. We found that patients with oral squamous cell carcinoma and a history of PVL (OSCC-PVL) were more often women compared to OSCC-noPVL patients. None of our OSCC-PVL patients smoked, while it was noted in OSCC-noPVL. Gingival localization was more frequent in OSCC-PVL, while tongue localization was significantly more common in OSCC-noPVL Erythroplastic clinical forms were more prevalent in OSCC-PVL, whereas ulcerated forms were more common in OSCC-noPVL. We also observed increased lymph node involvement in OSCC-noPVL, and the most advanced TNM stages were found in OSCC-noPVL.

Consequently, in the present study, we examined more cases in both groups: 50 with OSCC-PVL and 90 with OSCC-noPVL. With this increase, the previously detected differences became significant. For instance, 70% of the OSCC-PVL cases occurred in women versus 34.4% in the OSCC non-PVL group (χ^2^ = 116,307, *p* < 0.01), confirming the higher tendency for oral squamous cell carcinomas to develop in females with OSCC-PVL. This finding is similar to the 64.02% Ramos-Garcia et al. [15] reported for women with OSCC-PVL. In comparison, the frequency of tobacco consumption was not a significant factor in OSCC-PVL compared to OSCC-noPVL. Smoking was not a significant etiological factor in OSCC-PVL compared to OSCC-noPVL, where 34% were smokers and 66% were non-smokers in the OSCC-PVL (Table 1). Furthermore, the mean ages of the patients with OSCC-PVL and OSCC-noPVL were not significantly different.

The most common sites for conventional OSCC are the tongue (40%) and floor of the mouth (33%) [35]. Oliver et al. [36] found that the most frequent location of OSCC was on the tongue (28.26% of cases), followed by the floor of the mouth (26.09%). Mashberg et al. [37] studied 102 symptomatic cases of OSCC and the floor of the mouth, oral tongue, and soft palate complex accounted for 75% of all locations and 84% when the posterior pillar was excluded. The soft palate alone accounted for 75% of all locations, and 84% of locations when the posterior abutment was excluded. In contrast to the typical location of OSCC-noPVL, in our OSCC-PVL cases, the gingiva was the most frequently affected area, accounting for 50% of the cases (Table 1), while the most common locations in OSCC-noPVL were the tongue (41.4%) and gingiva (28.9%). This concurs with reports that the tongue was the most frequent location in OSCC-noPVL [35,38].

Although OSCC survival has not changed substantially in recent years, an early diagnosis is the most important [39]. According to González-Ruiz et al. [39], the primary cause of the high mortality rate in oral squamous cell carcinoma is its diagnosis at advanced stages (T3 and T4), where treatment often has poor efficacy, leading to challenges, mutilations, or disabilities. Amezaga-Fernandez et al. [26] reported that cases diagnosed at advanced stages accounted for 48.1% of their sample and that 38.7% relapsed. The 5-year OS and DSS rates were 39.9% and 46.1%, respectively [28]. González-Ruiz et al. [39] found that oral squamous cell carcinomas accompanied by PVL had a better prognosis than OSCC-noPVL. In this sense and in our current study we must highlight that 80% of our oral squamous cell carcinomas with PVL (OSCC-PVL) were diagnosed with a size that corresponded to a T1 or T2, therefore initial cases, while in the OSCC-noPVL group, were in T1 or T2 57.78% (*p* < 0.01). To our knowledge, this is the largest comparative clinical study between groups of patients with OSCC-PVL and OSCC-noPVL.

## 5. Limitations of This Study

The most important limitation of this study is that as PVL is a rare entity, our group of patients with OSCC-PVL could only consist of 50 cases.

Secondly, it is a retrospective study, and it would be ideal to conduct prospective comparative studies between the two groups. However, this would require follow-ups over many years, as PVL manifests very slowly and typically takes time to develop into oral squamous cell carcinomas. When they do arise, they often lead to the development of secondary primary tumors.

Third, this is solely a clinical study of oral lesions in OSCC, as indicated in the article’s title and objective in the introduction. We analyzed and compared the differential clinical characteristics between the two groups, including age, gender, location, clinical type, and tumor size, without considering the details of histologic findings, such as depth of invasion, perineural invasion, or lymphovascular invasion. This limited our focus to assessing only oral tumor size at the time of diagnosis without addressing histological aspects like the depth of invasion, which is a significant factor in the 8th edition of TNM.

## 6. Future Study Directions

In this current study, we have compared the clinical characteristics of two groups of oral squamous cell carcinoma patients (OSCC-PVL vs. OSCC-noPVL). Future directions will primarily focus on conducting prognostic and survival studies that compare both patient groups, examining the impact of histological findings on prognosis and the results of the various oncological treatments utilized: surgery, radiotherapy, and modern oncological therapies such as immunotherapy. 

## 7. Conclusions

In conclusion, for OSCC patients with PVL, the Bayesian logistic regression model revealed differences among the variables, stage, location, gender, and clinical type between groups that discriminated between OSCC-noPVL and OSCC-PVL patients. We found that OSCC preceded by PVL was much more frequent in women, had less aggressive clinical forms, and had significantly more frequent early T sizes than in OSCC-noPVL.

## Figures and Tables

**Figure 1 cancers-17-01199-f001:**
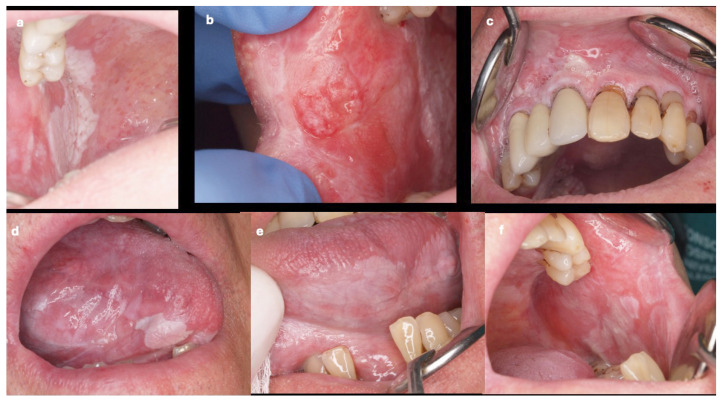
Case of proliferative verrucous leukoplakia. (**a**) leukoplakia lesions in the posterior third of the right buccal mucosa; (**b**) oral squamous cell carcinoma in initial erythroplastic form in the anterior third of the right buccal mucosa; (**c**) large leukoplakia lesions in the vestibular area of the upper gingiva; (**d**) leukoplakia lesions on the right lateral border of the tongue; (**e**) Leukoplakia on the left lateral border of the tongue; and (**f**) Leukoplakia on the left buccal mucosa. All these images (**a**–**f**) correspond to the same patient with OSCC-PVL.

**Figure 2 cancers-17-01199-f002:**
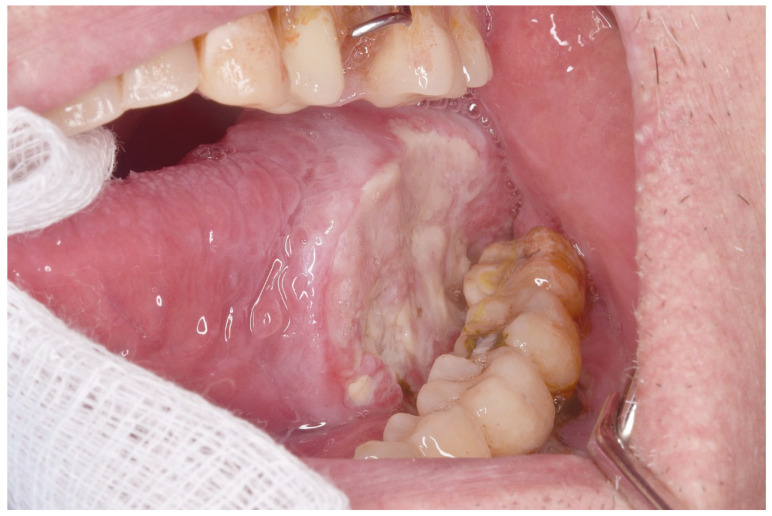
Advanced case of ulceration on the left lateral edge of the tongue with extensive infiltration. It is almost OSCC-non-PVL.

**Figure 3 cancers-17-01199-f003:**
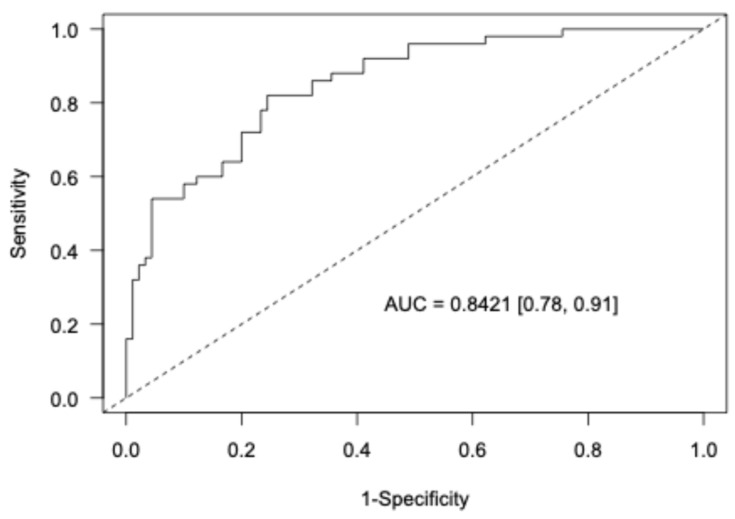
ROC curve depicting the discrimination power of the Bayesian logistic regression model.

**Table 1 cancers-17-01199-t001:** Clinical characteristics of the conventional oral squamous cell carcinoma and the PVL- oral squamous cell carcinoma patients.

Variable	OSSC-noPVL (*n* = 90)	PVL-OSCC (*n* = 50)	Statistical Significance
Age			
mean (sd)	69.51 (11.27)	66.62 (11.59)	*p* > 0.05
median (q1, q3)	70 (60.25, 77)	66 (59, 73.75)	
Gender			*p* < 0.001
Male	59 (65.56%)	15 (30%)	
Female	31 (34.44%)	35 (70%)	
Tobacco	50 (56.2%)	17 (34%)	*p* = 0.01
Location OSCC			*p* < 0.001
Gingiva	26 (28.89%)	25 (50%)	
Buccal mucosa	8 (8.89%)	8 (16%)	
Tongue	37 (41.11%)	9 (18%)	
Lips	2 (2.22%)	7 (14%)	
Floor of the mouth	6 (6.67%)	0 (0%)	
Palate	11 (12.22%)	1 (2%)	
Clinical.type			*p* = 0.003
Erytrholeukoplakia	7 (7.78%)	15 (30%)	
Ulcerative	57 (63.33%)	21 (42%)	
Exophitic	16 (17.78%)	11 (22%)	
Mixed	10 (11.11%)	3 (6%)	
T size			*p* = 0.008
T1 and T2	52 (57.78%)	40 (80%)	
T4 and T4	38 (42.22%)	10 (20%)	

OSSC-noPVL: Conventional Oral squamous cell carcinoma, without proliferative verrucous leukoplakia. OSSC-PVL: Oral squamous cell carcinoma with proliferative verrucous leukoplakia.

**Table 2 cancers-17-01199-t002:** Results of the Bayesian logistic regression model.

Variables	Estimate	Std. Error	OR	95% CrI	Prob. Effect
Intercept	0.432	0.757	-	[−1.1, 1.9]	0.711
Location buccal mucosa	0.865	0.712	2.376	[0.58, 10.0]	0.891
Location tongue	−1.16	0.546	0.313	[0.10, 10.0]	0.988
Location lips	1.028	0.951	2.797	[0.49, 21.7]	0.867
Location floor of the mouth	−3.496	1.799	0.03	[0.001, 0.64]	0.993
Location palate	−2.017	1.128	0.133	[0.01, 1.0]	0.974
T size advanced	−1.095	0.524	0.335	[0.11, 0.91]	0.985
Gender F	1.621	0.507	5.058	[1.9, 14.0]	1
Clinical type Ulcerative	−1.578	0.656	0.206	[0.06, 0.73]	0.993
Clinicaltype Exophitic	−1.047	0.733	0.351	[0.08, 1.46]	0.924
Clinical type Mixed	−1.916	0.961	0.147	[0.02, 0.90]	0.98

## Data Availability

No new data were created; the clinical data from this study are described in the tables provided in the document (Table 1 and Table 2).

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
