# Peer review of "Malignant Transformation of Proliferative Verrucous Leukoplakia: A Description of the Clinical Oral Characteristics of These Squamous Cell Carcinomas"

_cancers, 2025, doi:10.3390/cancers17071199_

Round 1

Reviewer 1 Report

Comments and Suggestions for Authors

This retrospective study has a large number of patients and is very well structured. It also refers to many previous studies. However, it is not an original study and the conclusions do not lead to anything new. Therefore, I do not think it can be published in its current form.

Author Response

REVIEWER 1

Comments 1: This retrospective study has a large number of patients and is very well structured. It also refers to many previous studies. However, it is not an original study, and the conclusions do not lead to anything new. Therefore, I do not think it can be published in its current form.

Response 1: Thank you for bringing this to our attention. The differences between oral squamous cell carcinomas in patients with Proliferative Verrucous Leukoplakia (PVL-OSCC) and conventional oral squamous cell carcinoma (OSCC) have indeed been documented previously; however, existing studies involve a limited number of cases (Herreros-Pomares A, Hervás D, Bagan-Debon L, Proaño A, Garcia D, Sandoval J, Bagan J. Oral cancers preceded by proliferative verrucous leukoplakia exhibit distinctive molecular features. Oral Dis. 2024 Apr;30(3):1072-1083. doi: 10.1111/odi.14550. Epub 2023 Mar 21. PMID: 36892444.) with only a few patients in both cancer types (8 PVL-OSCC and 10 conventional OSCC cases). Therefore, the above article is regarded as a preliminary investigation needing validation with a significantly larger patient cohort, which we have accomplished. Consequently, our current study involving 50 PVL-OSCC and 90 conventional OSCC cases supports and confirms that the preliminary findings are valid. Additionally, another group (González-Moles MÁ, Warnakulasuriya S, Ramos-García P. Prognosis parameters of oral carcinomas developed in proliferative verrucous leukoplakia: A systematic review and meta-analysis. Cancers (Basel). 2021 Sep 28;13(19):4843. doi: 10.3390/cancers13194843. PMID: 34638327; PMCID: PMC8507842.) has also emphasized these differences, although it was a systematic review and meta-analysis.
We greatly appreciate the comments from Reviewer 1. However, for the reasons stated above, we sincerely believe this clinical study contributes to the literature.

Reviewer 2 Report

Comments and Suggestions for Authors

The authors investigated a topic which is important for all the clinicians involved in oral cancer management, namely the characteristics of oral cancer arising from proliferative verroucous leukoplakia (PVL).

The design of the study is appropriate, although some clarifications are needed. 

The authors used the criteria of Cerero-Lapiedra et al. for classifying the PVL, this paper was published in 2010. On the other hand, the authors included patients via a retrospective charts analysis starting from 2005, how are they sure that those patients were actually PVL according to the Cerero-Lapiedra criteria?

The authors used a Bayesian multivariate regression model. Why did they choose this statistical analisis instead of the classical frequentist one?

They do not specifiy how did they test for normality for continuos variables and the results of the tests for normality.

The table should also indicate the T and N status and the respective p-values to test if any difference exist among the two groups. 

In general, the statistical anlyses should be describeb better and synthesized better in a single table together with all the variables. 

When referring to the malignant transformation of PVL they should cite the paper "Iocca O, Sollecito TP, Alawi F, et al. Potentially malignant disorders of the oral cavity and oral dysplasia: A systematic review and meta-analysis of malignant transformation rate by subtype. Head Neck. 2020 Mar;42(3):539-555. doi: 10.1002/hed.26006. Epub 2019 Dec 5. PMID: 31803979." which gives a thorough estimate of malignant transformation of PVL and other common potentially malignant disorders. 

The discussion should be shortened and should focus more on the results of the study and not just on the results of other papers (or at least these should be contextualized according to the results of their paper)

Author Response

REVIEWER 2

Comments 1: The design of the study is appropriate, although some clarifications are needed.

The authors used the criteria of Cerero-Lapiedra et al. for classifying the PVL, this paper was published in 2010. On the other hand, the authors included patients via a retrospective charts analysis starting from 2005, how are they sure that those patients were actually PVL according to the Cerero-Lapiedra criteria?

Response 1: Thank you for pointing this out. The criteria indicated by Cerero-Lapiedra et al. in 2010 (Cerero-Lapiedra R, Baladé-Martínez D, Moreno-López LA, Esparza-Gómez G, Bagán JV. Proliferative verrucous leukoplakia: a proposal for diagnostic criteria. Med Oral Patol Oral Cir Bucal. 2010 Nov 1;15(6):e839-45. PMID: 20173704.) consist of several major criteria and some minor criteria.

Major Criteria (MC):

  1. A leukoplakia lesion with more than two different oral sites, which is most frequently found in the gingiva, alveolar processes and palate.
  2. The existence of a verrucous area.
  3. That the lesions have spread or engrossed during development of the disease.
  4. That there has been a recurrence in a previously treated area.
  5. Histopathologically, there can be from simple epithelial hyperkeratosis to verrucous hyperplasia, verrucous carcinoma or oral squamous cell carcinoma, whether in situ or infiltrating.

Minor Criteria (mc):

  1. An oral leukoplakia lesion that occupies at least 3 cm when adding all the affected areas.
  2. That the patient be female.
  3. That the patient (male or female) be a non-smoker.
  4. A disease evolution higher than 5 years.

In order to make the diagnosis of PVL, it was suggested that one of the two following combinations of the criteria mentioned before were met.

  1. Three major criteria (being E among them) or
  2. Two major criteria (being E among them) + two minor criteria.

As seen from these major and minor criteria, all of them can and should be included in the patient's medical history. Therefore, in this retrospective study, it was possible to find the various points of these criteria in the records of the patients with PVL. Therefore, there were no problems in retrospectively diagnosing the patients.

Furthermore, we should note that PVL is often diagnosed retrospectively when a patient has been monitored over the years. We can refer to recent studies whose titles specifically mention the retrospective nature of the criteria, such as:

-Moreira MD, Maia FD, Zimbrão VL, Collodetti E, Grão-Velloso TR, Pimenta-Barros LA, Lourenço SQC, Camisasca DR. Demographic and clinicopathological comparison among oral lichen planus, lichenoid lesions and proliferative verrucous leukoplakia: a retrospective study. BMC Oral Health. 2024 Dec 19;24(1):1512. doi: 10.1186/s12903-024-05305-3. PMID: 39702188; PMCID: PMC11661046.

-Prabhu Venkatesh D, Ramalingam K, Ramani P, Krishnan M, Kumar Vadivel J. Epidemiological Trends and Clinicopathological Characteristics of Oral Leukoplakia: A Retrospective Analysis From a Single Institution in Chennai, Tamil Nadu, India. Cureus. June 3, 2024;16(6):e61590. doi: 10.7759/cureus.61590. PMID: 38962636; PMCID: PMC11221497.

Comment 2: The authors used a Bayesian multivariate regression model. Why did they choose this statistical analysis instead of the classical frequentist one?

They do not specify how they tested for normality for continuous variables and the results of the tests for normality.

Response 2: Thank you for your comment. The motivation for employing a Bayesian multivariable regression arises from its ability to integrate regularizing priors when estimating the effects of various variables, thereby enhancing the reliability and robustness of the results despite the limited sample size (see, for example, Gelman A, Jakulin A, Pittau MG, Su YS. A weakly informative default prior distribution for logistic and other regression models). We have added a sentence in the methods section to emphasize the rationale for utilizing this methodology. Regarding the normality tests, we apologize for the confusion. No normality test was conducted since the only numerical variable was age, and we compared its values between groups using the Wilcoxon-Mann-Whitney test. Therefore, normality was not assumed. We have removed the mention of the normality test in the methods section. Additionally, we have rewritten the methods section to enhance its readability.

Changes in the manuscript (Materials and Method section):

“The data were summarized using the mean (standard deviation) and median (1st and 3rd quartiles) for numerical variables and absolute (relative) frequencies for categorical variables.

We compared quantitative and qualitative variables between both groups to identify significant differences We used the Wilcoxon-Mann-Whitney test to compare quantitative variables. For qualitative variables, we performed a contingency analysis using the Chi-squared test. For these analyses we set the level of significance at p < 0.05.

Finally, a Bayesian multivariable logistic regression model was adjusted to assess whether the different clinical variables discriminated between the two groups. The uncertainty of the estimates was assessed by estimating the 95% credible intervals, and the probability of direction (pd) was used as an index of the existence of the effect. Regularizing priors, N(0, 3) were used to improve the robustness and reliability of our model results. The performance of the model was assessed by estimating the area under the receiver operating characteristic curve (AUROC). Internal validation of the AUROC was performed using 10-fold cross-validation. All statistical analyses were performed using R (ver. 4.4.1) and the R packages brms (ver. 2.22.0) and bayestestR (ver. 0.15.0).”

Comment 3: The table should also indicate the T and N status and the respective p-values to test if any difference exist among the two groups.

Response 3: Thank you for pointing this out. As outlined in the title and objectives mentioned in the introduction, this study is strictly clinical. We analyze the patients' age, gender, and various clinical characteristics of the oral lesions in both cancer groups. These characteristics include the location of the oral squamous cell carcinoma (OSCC), the clinical type, and the size of the oral lesions as defined by the T value of the TNM classification. We distinguish between two subgroups: initial sizes in cancer (T1 and T2) and advanced sizes (T3 and T4), effectively categorizing them. We do not assess the histological aspects of the cancer lesions; this is purely a clinical study, nor is it a prognostic-evolutionary study of either type of OSCC. We added the p-values in each group, as requested by Reviewer 2.

Comment 4: When referring to the malignant transformation of PVL they should cite the paper "Iocca O, Sollecito TP, Alawi F, et al. Potentially malignant disorders of the oral cavity and oral dysplasia: A systematic review and meta-analysis of malignant transformation rate by subtype. Head Neck. 2020 Mar;42(3):539- 555. doi: 10.1002/hed.26006. Epub 2019 Dec 5. PMID: 31803979." which gives a thorough estimate of malignant transformation of PVL and other common potentially malignant disorders.

Response 4: Thank you for pointing this out. We agree with Reviewer 2 and have included the bibliographic reference they suggested in the discussion section.

Comment 5: The discussion should be shortened and should focus more on the results of the study and not just on the results of other papers (or at least these should be contextualized according to the results of their paper)

Response 5: Thank you for pointing this out. According to Reviewer 2, we have significantly reduced the discussion and concentrated more on the study's results.

Reviewer 3 Report

Comments and Suggestions for Authors

Interesting article on a topic with clinical relevance. However, there are some aspects that need improvement.

  • In Table 1, the p-values from the statistical analysis should be included; The term "cancer" should be replaced with "OSCC" to maintain consistency with the text.
  • In the captions of the two figures on page 4, the numbering should be corrected.
  • The discussion is too long and covers topics unrelated to the study's objective. The entire discussion should be revised, as only the final part (from line 232 to line 282) is somewhat relevant to the study.
  • Lines 266-270 mention data on patients' smoking habits, but this parameter was not assessed in the Methods & Materials section or in the results.
  • The selection of references should be more rigorous, as there are too many self-citations.
  • The conclusion must also be revised, as it is incorrect and differs significantly from the one in the abstract.

Author Response

REVIEWER 3

Comment 1:

The p-values from the statistical analysis should be included in Table 1. To maintain consistency with the text, the term "cancer" should be replaced with "OSCC.”

Response 1: Thank you for pointing this out. In Table 1, we have included the p-values. To ensure consistency, as noted by Reviewer 3, we have replaced the term “ cancer “ with “ oral squamous cell carcinoma (OSCC). “

Comment 2:

In the captions of the two figures on page 4 the numbering should be corrected.

Response 2: Indeed, as Reviewer 3 noted, there was an error in the captions of the two figures. We have fixed it in the final document.

Comment 3:

The discussion is too long and covers topics unrelated to the study's objective. The entire discussion should be revised, as only the final part (from line 232 to line 282) is somewhat relevant to the study.

Response 3: Thank you for pointing this out. As we have replied to reviewer 2, who mentioned the same issue regarding the need to reduce the discussion, we have made the appropriate adjustments, and we fully agree. We have concentrated on what is pertinent to the study.

Comment 4:

Lines 266-270 mention data on patients' smoking habits, but this parameter was not assessed in the Methods & Materials section or the results.

Response 4: Indeed, you are right. We did not initially refer to smoking habits in patients in Materials and Methods or Results.

We have not done so because numerous publications demonstrate that tobacco is not an etiological factor in PVL, unlike conventional OSCC. For this reason, tobacco is not associated with PVL. Below are some references indicating this lack of relationship between tobacco and PVL, which is why we did not initially include it as another variable.

For example, this year, 2025, Mohideen et al. (Mohideen K, Ghosh S, Krithika C, Ali-Hassan M, Chole R, Dhungel S. Malignant transformation of proliferative Verrucous Leukoplakia-systematic review & meta-analysis. BMC Oral Health. Feb 1, 2025;25(1):175. doi: 10.1186/s12903-025-05565-7. PMID: 39893387; PMCID: PMC11786437.) say the following: Tobacco and alcohol showed weak associations, with most patients being non-smokers (59.54%) and non-drinkers (78.18%)

Ramos-Garcia et al. (Malignant transformation of oral proliferative verrucous leukoplakia: A systematic review and meta-analysis. Oral Dis. 2021 Nov;27(8):1896-1907. doi: 10.1111/odi.13831. Epub 2021 May 19. PMID: 34009718.) indicate that no conclusive results were found between malignant transformation and sex or age distribution, tobacco, or alcohol consumption in proliferative verrucous leukoplakia.

Despite this, and as requested by Reviewer 3, we have now included the variable of tobacco consumption in Table 1. As anticipated, tobacco consumption is statistically and significantly lower in the PVL group, as many authors have previously noted.

Comment 5: The selection of references should be more rigorous, as there are too many self-citations.

Response 5: Thank you for pointing this to our attention. We sincerely believe that we have been rigorous because everything we cited comes from high-impact journals (included in the Journal Citation Report), where we are authors. These correspond to articles where we felt we contributed to the literature but did not intend to be uncritical of our self-citation. As a result, we have decided to remove some of our publications from the references, as suggested by Reviewer 3. The bibliography in the manuscript has been revised to remove some of our citations. There were 51 bibliographic citations in the initial manuscript; now, we have reduced it to 39, eliminating several of our previously cited references. In the initial submission of the 51, there were 11 articles with authorship of our research group and now we have reduced to 5 that we have left, which we consider necessary.

Comment 6: The conclusion must also be revised as it is incorrect and differs significantly from the one in the abstract.

Response 6: Thank you for pointing this out. We apologize because reviewer 3 is correct in his/her observations. We will redo the conclusions according to what is indicated in the abstract.

Reviewer 4 Report

Comments and Suggestions for Authors

This is an interesting and valid report on a poorly understood condition. The paper should be enhanced by at least including data on the staging , I could only find the T stage. The N and M stages should be reported even if there were no N+ or M+ cases. Additionally, it would be of benefit to know the recurrence and survival status of the 2 groups . A report of the treatment initiated for the 2 groups and pathological outcomes of any surgical procedures would further enhance the paper and guide the reader with regard to appropriate management of this condition.

The discussion can be shortened by removing the discussion on unrelated cancers, i.e. kidney cancers line174.

Author Response

REVIEWER 4

Comment 1: This is an interesting and valid report on a poorly understood condition. The paper should be enhanced by at least including data on the staging , I could only find the T stage. The N and M stages should be reported even if there were no N+ or M+ cases. Additionally, it would be of benefit to know the recurrence and survival status of the 2 groups . A report of the treatment initiated for the 2 groups and pathological outcomes of any surgical procedures would further enhance the paper and guide the reader with regard to the appropriate management of this condition.

Response 1: Thank you for pointing this out. Reviewer 2 in question 3, mentioned that the table should also indicate the T and N status and the respective p-values to test whether any difference exists among the two groups. We already replied to him/her that in Table 1, we have added the values of T and N with the respective p-values to test the differences between the two groups.

However, we must emphasize that the primary objective of this study was to clinically compare the findings of tumors within the oral cavity, rather than conducting a prognostic or survival study that would determine the therapeutic modality based on TNM classification and its correlation with subsequent survival. We address this within the limitations of the study, as noted at the end of the discussion in the article. Consequently, we concentrated on the T values, indicating whether the cases were classified as initial or advanced based on the tumor's T value.

We want to point out that the article title is “Malignant transformation of proliferative verrucous leukoplakia: A description of the clinical oral characteristics of these oral squamous cell carcinomas”. In other words, as we indicated in the last paragraph of the introduction, the objective of this work was very specific: The current study compares the “clinical characteristics” of large groups with OSCC-PVL and OSCC-noPVL, to validate our preliminary results and demonstrate that OSCC-PVL has different clinical behavior to that of OSCC-noPVL.

In this study, we did not set out to analyze the response to treatments or carry out a survival and prognosis analysis of these patients. That would be a matter for another investigation, in our view. Therefore, we mainly want to highlight the clinical aspects of our two groups, such as gender, locations, clinical types, and whether the cancer was in an early or late T size.

Comment 2: The discussion can be shortened by removing the discussion on non-related cancers, i.e., the kidney cancers line 174.

Response 2: Thank you for bringing this to our attention. We agree with Reviewer 4 that the discussion should be reduced. This is consistent with our responses to Reviewers 2 and 3. Therefore, we fully support your suggestion. In the final document, we have significantly reduced the discussion by primarily focusing on analyzing our results.

Reviewer 5 Report

Comments and Suggestions for Authors

Dear Authors,

This study aims to contribute to dental research by comparing the clinical characteristics of patients with oral squamous cell carcinoma (OSCC) who had and had not been previously diagnosed with PVL.

This manuscript's subject is not only interesting but also of great value because, as it emphasizes, proliferative verrucous leukoplakia (PVL) has a high rate of transformation into oral cancer, and OSCC-PVL has better clinical outcomes and more favorable prognoses than OSCC-noPVL, particularly in terms of mortality rates. Your work in this field is highly appreciated. However, some concerns related to the report need to be addressed.

Introduction

While a good start, the Introduction could benefit from providing more comprehensive information related to the topic. This will establish the subject's current status and the potential contribution of your work to the field.

  • More studies and information regarding the malignant transformation of PVL will strengthen the rationale for the present research. Information about the clinical characteristics of OSCC will also provide a solid base for this study.
  • Avoid redundancy, some information is presented repetitively in this section and discussions.
  • The study's aim is mentioned in the introduction. Still, it is unclear whether the focus is on the clinical characterization of OSCC-PVL or the prognostic differentiation from OSCC-noPVL.

Materials and Methods

In this section, all steps should be explained clearly and in detail, and enough information should be provided so that another researcher can easily replicate the study:

  • How was the sample size calculated?
  • It would be helpful to describe the inclusion/exclusion criteria in detail and provide more information about the study design.
  • No details are mentioned about the history of alcohol and tobacco consumption in the patients analyzed. However, they are major risk factors for OSCC, although, in the discussion section, reference is made to Table 1 when talking about smoking. Still, this information is not presented in this table.

The statistical analysis is well presented, but:

  • The study uses Bayesian logistic regression, but it is unclear why this method was chosen over a classical approach (e.g., standard multivariate logistic regression).
  • It would be advisable to provide information on the used normality tests
  • Whether a statistical power analysis was performed to determine the minimum sample size required to detect significant differences is not specified.
  • It is unclear whether the model was externally validated with an independent data set.

Results

  • It should not duplicate the information presented in the text with those in the tables.

Although appropriate statistical methods are used and the results are clearly presented, certain checks and interpretations are missing that could significantly improve the clarity and robustness of the conclusions.

  • Association measures or dispersion indicators should be provided for clinical variables to better understand the independent effect of each factor.
  • Correlation coefficients for relationships between categorical variables

Discussion

  • The discussion spends too much space summarizing global cancer incidence trends, irrelevant to the study's core research question. Instead, it would have been more useful to focus on OSCC-PVL-specific epidemiological trends.
  • The discussion focuses more on a previous study and less on the results of the present study. I suggest that you extend the discussion to the present study and interpret the results in depth. Also, as I said above, this section contains information about patients who are tobacco users, but neither the materials and methods section nor the results section refers to this aspect.
  • Acknowledge the study's limitations more explicitly, and future study directions should be presented clearly.

Conclusions

  • Must be presented in accordance with the study's limitations
  • The last statement is not a conclusion; I suggest you remove it; it fits better in the previous section.

Author Response

REVIEWER 5

Comment 1: Introduction

While a good start, the Introduction could benefit from more comprehensive information related to the topic. This will establish the subject's current status and the potential contribution of your work to the field.

More studies and information regarding the malignant transformation of PVL will strengthen the rationale for the present research. Information about the clinical characteristics of OSCC will also provide a solid base for this study.

Response 1: Thank you for bringing this to our attention. We agree with this comment. In alignment with Reviewer 5, we have included additional articles in the introduction that address the malignant transformation of Proliferative Verrucous Leukoplakia (PVL) and discuss the clinical characteristics of the resulting oral squamous cell carcinomas (OSCC). We aim to avoid redundancy in the discussion section, as we will also compare our findings with those of other authors. However, it is important to note that there are few articles presenting case series on malignant transformation; instead, systematic reviews and meta-analyses are more prevalent.

Comment 2: Introduction

The study's aim is mentioned in the introduction. Still, it is unclear whether the focus is on the clinical characterization of OSCC-PVL or the prognostic differentiation from OSCC-noPVL.

Response 2: Thank you for pointing this out. As we have replied to reviewer 4, we would like to point out that the article title is “Malignant transformation of proliferative verrucous leukoplakia: description of the clinical oral characteristics of these oral squamous cell carcinomas”. In other words, as we indicated in the last paragraph of the introduction, the objective of this work was very specific: The present study compares the clinical characteristics of large groups with OSCC-PVL and OSCC-non-PVL, to validate our preliminary results and demonstrate that OSCC-PVL has a different clinical behavior than OSCC-non-PVL.

In this study, we did not set out to analyze the response to treatments or to carry out a survival and prognosis analysis in these patients. That would be a matter for another piece of research. Therefore, we mainly want to highlight the clinical aspects of our two groups, such as sex, location, clinical types, and whether it was an early or late stage of cancer.

Following reviewer 5 and to avoid the doubts he indicates, we will eliminate the aspects that could have been included or discussed at the prognostic level in the discussion and focus only on the clinical characteristics.

Comment 3: Materials and Methods

How was the sample size calculated?

Response 3: Thank you for raising this important question. As stated in the methods section, this study is retrospective, so we included all available samples for the OSCC-PVL group. Regarding the number of samples for the OSCC-noPVL group, a power analysis was performed using Monte Carlo simulations to assess the best ratio of OSCC-noPVL to OSCC-PVL samples that would improve the statistical power without dramatically increasing the sample size. The simulation results concluded that a 2:1 ratio was the most cost-effective approach.

Comment 4: Materials and Methods

It would be helpful to describe the inclusion/exclusion criteria in detail and provide more information about the study design.

Response 4: Following reviewer 5, we have included Cerero-Lapiedra's inclusion criteria, indicating the requirements for considering it a PVL. We have also expanded and detailed the inclusion criteria section in the materials and methods and highlighted it in the new Word document.

Comment 5: Materials and Methods

No details are mentioned about the history of alcohol and tobacco consumption in the patients analyzed. However, they are major risk factors for OSCC, although, in the discussion section, reference is made to Table 1 when talking about smoking. Still, this information is not presented in this table.

Response 5: Just as I answered reviewer 3 to this same question, I will answer reviewer 5:

You are right. We did not initially refer to smoking habits in patients in Materials and Methods or Results.

We did not do so because numerous publications have shown that tobacco and alcohol are not etiological factors in PVL, unlike conventional OSCC. Below are some references that indicate this lack of relationship between tobacco and PVL, so we did not initially include it as another variable.

For example, in 2025, Mohideen et al. (Mohideen K, Ghosh S, Krithika C, Ali-Hassan M, Chole R, Dhungel S. Malignant transformation of proliferative verrucous leukoplakia: systematic review and meta-analysis. BMC Oral Health. February 1, 2025; 25(1):175. doi: 10.1186/s12903-025-05565-7. PMID: 39893387; PMCID: PMC11786437.) state the following: Tobacco and alcohol showed weak associations, as the majority of patients did not smoke (59.54%) or drink (78.18%) in PVL.

Ramos-García et al. (Malignant transformation of oral proliferative verrucous leukoplakia: a systematic review and meta-analysis. Oral Dis. 2021 Nov;27(8):1896-1907. doi: 10.1111/odi.13831. Epub 2021 May 19. PMID: 34009718.) indicate that no conclusive results were found between malignant transformation (MT) and gender or age distribution, smoking, or alcohol consumption in proliferative verrucous leukoplakia.

Despite this, and at the authors' request, we have included the variable of tobacco consumption in the new Table 1. As anticipated, tobacco consumption is statistically and significantly lower in the PVL group, as many authors have previously noted.

Comment 6: Materials and Methods

The statistical analysis is well presented, but:

The study uses Bayesian logistic regression, but it is unclear why this method was chosen over a classical approach (e.g., standard multivariate logistic regression).

It would be advisable to provide information on the used normality tests

Whether a statistical power analysis was performed to determine the minimum sample size required to detect significant differences is not specified.

It is unclear whether the model was externally validated with an independent data set.

Response 6: Thank you for your comment. As explained to another reviewer who raised a similar concern, the motivation for using a Bayesian model was the ability to apply regularizing priors for estimating the effects of the different variables, thereby making the results more reliable and robust despite the limited sample size (see, for example, Gelman A, Jakulin A, Pittau MG, Su YS. A weakly informative default prior distribution for logistic and other regression models.). We have added a sentence to the methods section to emphasize the motivation behind using this methodology.

We apologize for the confusion regarding the normality tests. No normality test was conducted because the only numerical variable was age, which had its values compared between groups using the Wilcoxon-Mann-Whitney test; thus, normality was not assumed. Consequently, we have removed the mention of the normality test from the methods section.

Statistical power analysis was performed to determine the ratio of OSCC-noPVL to OSCC-PVL samples that were most cost-effective. However, since this was a retrospective study, the number of OSCC-PVL samples was determined by their availability.

As stated in the methods section, the model was validated using 10-fold cross-validation, so no external validation was performed. Such validation would have been extremely useful, but it was out of scope for this retrospective study.

Comment 7: Results

It should not duplicate the information presented in the text with those in the tables.

Response 7: As noted by Reviewer 5, we have revised the results section to avoid repetition with the tables.

Comment 8: Results

Although appropriate statistical methods are used and the results are clearly presented, certain checks and interpretations are missing that could significantly improve the clarity and robustness of the conclusions.

Association measures or dispersion indicators should be provided for clinical variables to better understand the independent effect of each factor.

Correlation coefficients for relationships between categorical variables

Response 8:

Thank you for this observation. Association measures are presented in Table 2, as the odds ratios (OR) and their corresponding 95% credible intervals are provided for the independent effect of each factor.

We don’t understand what is meant by “correlation coefficients for relationships between categorical variables. " The relationship between categorical variables was assessed using Chi-squared tests, as stated in the methods section.

Comment 9: Discussion

The discussion spends too much space summarizing global cancer incidence trends, irrelevant to the study's core research question. Instead, it would have been more useful to focus onOSCC-PVL-specific epidemiological trends.

The discussion focuses more on a previous study and less on the results of the present study. I suggest you extend the discussion to the present study and interpret the results in depth.

Response 9: Thank you for highlighting this. As we explained to reviewers 2, 3, and 4 regarding their suggestion to reduce the discussion, we affirmed to reviewer 5 that we fully agree. We appreciate his recommendation to streamline the discussion, focusing primarily on aspects related to our results. 

Comment 10: Discussion

Also, as I said above, this section contains information about patients who are tobacco users, but neither the materials and methods section nor the results section refers to this aspect.

Response 10: In the same way we replied to reviewer 3, we responded to reviewer 5. Indeed, you are right; we did not initially refer to smoking habits in patients in Materials and Methods or Results.

We have not done so because numerous publications have shown that tobacco is not an etiological factor in PVL, unlike conventional OSCC. Below are some references that illustrate this lack of relationship between tobacco and PVL, which is why we did not initially include it as another variable.

For example, this year, 2025, Mohideen et al. (Mohideen K, Ghosh S, Krithika C, Ali-Hassan M, Chole R, Dhungel S. Malignant transformation of proliferative verrucous leukoplakia- systematic review & meta-analysis. BMC Oral Health. Feb 1, 2025;25(1):175. doi: 10.1186/s12903-025-05565-7. PMID: 39893387; PMCID: PMC11786437) state the following: Tobacco and alcohol exhibited weak associations, with most patients being non-smokers (59.54%) and non-drinkers (78.18%) in PVL.

Ramos-Garcia et al. (Malignant transformation of oral proliferative verrucous leukoplakia: A systematic review and meta-analysis. Oral Dis. 2021 Nov;27(8):1896-1907. doi: 10.1111/odi.13831. Epub 2021 May 19. PMID: 34009718.) indicate that no conclusive results were found regarding malignant transformation (MT) in relation to gender or age distribution, as well as tobacco or alcohol consumption in proliferative verrucous leukoplakia.

Despite this, and as requested by the authors, we have now added the variable of tobacco consumption to Table 1. As expected, tobacco consumption is statistically and significantly lower in the PVL group, as numerous authors had previously pointed out.

Comment 11: Discussion

Acknowledge the study's limitations more explicitly, and future study directions should be presented clearly.

Response 11: Thank you for highlighting this. In line with the recommendation of Reviewer 5, we have added a section on limitations and future research directions at the end.

Comment 12: Conclusions

The last statement is not a conclusion; I suggest you remove it; it fits better in the previous section.

Response 12: Thank you for pointing this out. We have moved the last statement to the previous section, as suggested.

Round 2

Reviewer 2 Report

Comments and Suggestions for Authors

I am satisified by the answers provided by the authors

Author Response

Reviewer 2:

Comments 1: I am satisfied with the responses provided by the authors.

Response 1: Many thanks to Reviewer 2 for their comments, which initially led us to improve the article substantially.

We don't have to respond to anything else because Reviewer 2 has indicated that he/she is satisfied with our responses and has not asked us to address any other issues.

Reviewer 4 Report

Comments and Suggestions for Authors

Th authors should consider modification of the discussion line 221-225. As the article is about oral cancer the data provided that includes pharyngeal sites should be revised to remove the pharyngeal data and only include oral cancer.

Author Response

Reviewer 4:

Comments 1: The authors should consider modification of the discussion line 221-225. As the article is about oral cancer the data provided that includes pharyngeal sites should be revised to remove the pharyngeal data and only include oral cancer.

Response 1: Many thanks to Reviewer 4 for their comments.

In response to Reviewer 4, we have replaced the sentence referring to oral and pharyngeal cancers with another quote that focuses solely on oral cancers, sourced from a recent study article. Additionally, we updated the previous reference to the new one, reference 25.

The sentence we added in the discussion, after deleting the one that had pharyngeal cancers, is as follows:

Oral cancer represents nearly 3% of new cancer cases in the United States, with incidence rates rising over the past decade. The overall incidence of lip and oral cavity cancers is approximately 4.1 cases per 100,000 people; however, there is significant variation worldwide, with higher rates observed in Asian countries.

Reviewer 5 Report

Comments and Suggestions for Authors

Thank you to the authors for addressing all the raised issues.

Author Response

Reviewer 5:

Comments 1: Thank you to the authors for addressing all the raised issues.

Response 1: Many thanks to Reviewer 5 for accepting all our responses to his/her questions. We will not reply further since he indicated we have addressed all the issues.